# Effects of Superior–Subordinate Guanxi on Employee Innovative Behavior: The Role of Follower Dependency and Work Passion

**DOI:** 10.3390/bs13080645

**Published:** 2023-08-02

**Authors:** Zhiyong Han, Ming Ni, Chenbin Hou, Yuliang Zhang

**Affiliations:** School of Business Administration, Anhui University of Finance and Economics, Bengbu 233030, China; hzyong@aufe.edu.cn (Z.H.); 3202100351@aufe.edu.cn (M.N.); wave2021923@163.com (C.H.)

**Keywords:** superior–subordinate guanxi, follower dependency, work passion, employee innovative behavior

## Abstract

In the face of an increasingly complex competitive environment, a company’s ability to innovate is the key to a gaining sustainable competitive advantage. As the cornerstone of an organization’s survival and development, employee innovative behavior is key to enhancing an organization’s innovative capability. Based on a conservation of resources, this study investigates the mechanism of the role of superior–subordinate guanxi on employees’ innovative behavior from both emotional and cognitive perspectives. Through the analysis of 305 full-time employees’ research data, the results show that superior–subordinate guanxi can promote employees’ innovative behavior by stimulating their work passion and reducing their dependence on superiors. Our research provides certain theoretical guidance and policy recommendations for enterprises to improve the innovation ability of employees by revealing the internal mechanism of superior–subordinate guanxi affecting the innovation behavior of employees.

## 1. Introduction

Innovation is the key driver of enterprise development and the key factor for the perpetual vitality of a country. In the increasingly competitive market, sustainable innovation has become not only the core of enhancing the competitiveness of enterprises, but even the lifeblood of the entire national development. Therefore, the question of how to improve employees’ ability to innovate has become very important in various sectors of society [1].

Employee innovative behavior (EIB) refers to the behavior of employees putting new ideas, concepts, and problem-solving solutions into practice in their daily work and life [2]. Many leadership variables have been recognized as important influencing factors on creativity and innovation [3]. At this stage, many scholars aim to focus on the mechanism of leadership styles on EIB; for example, studies have found that inclusive, transformational leadership and responsible leadership can positively influence EIB [4,5,6]. However, few studies have focused on the effects of superior–subordinate guanxi (SSG), which is rooted in the Chinese cultural context, on EIB. Most scholars choose to use leader–member exchange (LMX) instead of SSG, but LMX, originating from Western culture, cannot adequately reflect the background of China’s “authority-oriented” and “relationship-oriented” culture [7]. Therefore, studying how SSG affects EIB is more in line with the cultural context of China. Guanxi, considered to be a construct rooted in the Chinese cultural context [8], is established and driven by personal interests and needs, either because of birth and blood, or because of social interactions. Reciprocal exchanges over time can maintain and strengthen guanxi, while at the same time, relationships can be transferred through certain mediums [9]. As an expression of relationships, SSG refers to a type of interpersonal connection that is constructed based on work, interests, emotions, and needs during a specific period of time. It includes four core elements: obligation, emotion, formal exchange, and human exchange [10]. Different qualities of SSG can have different impacts on employees’ emotions, psychology, and behavior [11]. Existing research explores the mediating effects of psychological empowerment, job satisfaction, and organizational commitment between SSG and EIB from the perspectives of social exchange and social cognition [7,12]. It is worth noting that scholars mostly discuss the mechanism of SSG on EIB from a cognitive perspective. However, individual behavior is not only driven by the “cold processing system of rational cognition”, but is also affected by the “hot processing system of emotional impulse” [13]. Therefore, this study suggests that the impact mechanism of SSG on EIB can be systematically explored from both cognitive and emotional perspectives.

Conservation of Resources (COR) theory holds that humans strive to protect their existing valuable resources. When an individual has enough resources, he/she will acquire further resources, and when his/her resources suffer and cannot be recovered for quite some time, the individual will reduce the resources in order to maintain the status quo [14]. SSG, as an important resource of relationships, can have a certain impact on EIB through both emotional and cognitive pathways. Firstly, EIB carries strong risks, requiring high recognition of their work value and a positive emotional state to complete the activity. Work passion itself is a positive emotional state, and it can help employees complete high-risk activities with a positive attitude [15,16]. Therefore, high-quality SSG increases employees’ emotional resources and strengthens their identification with their work, thereby increasing their likelihood of implementing innovative behaviors. Meanwhile, studies have shown that follower dependency can inhibit the development of employee creativity [17], and positive leadership styles, such as inclusive leadership [17] and responsible leadership [18], have been shown to increase follower dependency on leadership. SSG is different from the static leadership style as it emphasizes the interpersonal interaction between superiors and subordinates, and, under the condition of a close SSG, the leader will allocate more organizational resources to the subordinates with the closest relationship around them [19], which are not only beneficial to the career development of subordinates, but also help employees cope with pressure [20]. Employees, therefore, have a certain degree of autonomy and decision-making power and accept and share the organization’s values [21]. Employees who are no longer dependent on their leaders emphasize the pursuit of self and independence, can break the original work pattern, and are thus able to generate more new ideas at work and put them into practice. Based on this, our study suggests that a good SSG can stimulate employees to implement innovative behavior by increasing their work passion and reducing their dependence on superiors.

In summary, the goal of this study is to systematically explore the mechanism of action between SSG and EIB from cognitive and affective perspectives based on COR. By analyzing a sample of 305 full-time employees in China, a theoretical model was constructed and then tested in order to examine the impact of SSG on EIB (Figure 1). This study makes two main theoretical contributions. Firstly, based on COR, follower dependency and work passion are included in the study of the relationship between SSG and EIB, thereby enriching the research on the impact mechanism between the two. Secondly, unlike previous studies on follower dependency, this study is based on the belief that SSG reduces follower dependency in Chinese cultural contexts, enriching the research on the antecedents of follower dependency.

## 2. Literature Review and Hypothesis Development

### 2.1. SSG and EIB

In the Chinese context, SSG refers to the informal, idiosyncratic social connection between leaders and subordinates through interactive behaviors outside the confines of the workplace [22], and, unlike LMX, SSG is not only sustained in the workplace but is also manifested in out-of-work scenarios, such as in social life [10]. The SSG, as a relational interaction between leaders and subordinates, can have a profound impact on employees’ attitudes and behaviors [23]. Some studies have confirmed the positive impact of SSG on proactive behaviors, such as organizational citizenship behavior [24], and voice behavior [25]. EIB is the initiative of employees to motivate themselves to solve complex situations and problems in order to the innovative performance of their individual, team, or organization [26]. Psychological safety [27], perceived organizational support [28], and psychological empowerment [29] have all been identified as important influences on EIB.

This study argues that good SSG can promote EIB. First of all, good SSG provides employees with resource security [8], and they can better establish, expand, and utilize their network resources in order to obtain the kinds of resources that are needed for innovation. In addition to this, high-quality SSG can bring a certain sense of psychological security to employees [30], enabling employees to engage in risky activities, such as innovative work behaviors. Secondly, good SSG can be developed into acquaintance or even kinship relationships with characteristics such as trust, fondness, and respect and it is on this basis that subordinates will have a greater degree of autonomy and decision-making power. As a result, employees can have more resources, encouragement, and courage, all of which have been proven to stimulate the innovative behavior of employees [31]. Lastly, empirical studies have shown that high-quality superior–subordinate relationships can stimulate employees’ innovative behaviors by increasing their job satisfaction, their sense of psychological empowerment, and their commitment to the organization [7,12]. Based on the analysis above, we propose the following hypothesis:

**H1.** 
*SSG positively predicts EIB.*


### 2.2. SSG and Follower Dependency

Follower dependency is also known as dependence on leadership [32]. Birtchnell et al. (1988) [33] argue that a dependent person needs to receive recognition, guidance, and guidance from others in order to make up for their shortcomings and, more importantly, to accept, endorse, and advocate for the values of others. Follower dependency can be divided into cognitive dependency and motivational dependency. Follower cognitive dependency refers to the dependence of employees on leaders in information processing activities, such as sensation, perception, memory, imagination, and thinking, whilst motivational dependency refers to the dependence of employees on their leaders while maintaining a state of hard work [17]. Research has shown that leaders, as owners of organizational power and the controllers of various resources, must rely on their subordinates to ensure the effective implementation of work tasks and to achieve work goals. By establishing good relationships with leaders, employees can obtain leadership support and performance feedback as well as recognition, praise, and trust from their leaders [34].

Unlike previous studies, we consider that good SSG can reduce follower dependency. In China, the new generation of employees born in the 1980s and 1990s has been the main force in the labor market, and unlike the older generation, the new generation of employees dare to think and act, dare to pursue the self, and dare to disregard authority [35], which means that they pursue independence in thought and behavior and also reduce their dependence on superiors [36], and good SSG can thereby fulfill that pursuit of self. On the one hand, high-quality SSG implies that employees have access to more relational resources, and if this is the case, subordinates can satisfy the need to evince their own abilities and to exert autonomy [37], where subordinates have the opportunity to pursue their selves and reduce their dependence on their leaders. On the other hand, high-quality SSG emphasizes that superiors should respect the wishes of subordinates and treat subordinates as equals, which further enhances employees’ sense of ownership and creates a humanistic and caring organizational atmosphere [17] within the organization, which, in turn, increases the motivation of subordinates to pursue independence and emboldens them to discard dependence on their leaders. Therefore, we propose the following hypothesis:

**H2.** 
*SSG is negatively correlated with follower dependency.*


### 2.3. The Mediating Role of Follower Dependency

Employees with high dependency often uncritically accept, agree with, and advocate for the views of their superiors [33], while creativity, conversely, generates novel ideas [38]. Therefore, dependency and creativity clearly go against each other. Firstly, from a cognitive perspective, the strong admiration and attachment of subordinates to their leaders can easily lead them to fully accept their leaders’ viewpoints without any criticism, and to be unconditionally loyal to their leaders in matters of cognition [39], and this results in subordinates not actively proposing new ideas but, instead, merely adopting established solutions and established thinking patterns [40]. Secondly, from a motivational perspective, dependence can lead subordinates to seek praise and recognition from their leaders [41] and they may not openly express critical opinions that could lead to alienation from their leaders [40]. When a leader is absent, subordinates may then feel disoriented, thereby reducing engagement, work consciousness, and creativity [42]. 

This study argues that SSG can stimulate EIB by reducing follower dependency. Specifically, based on COR, close SSG is an important social resource for employees, and employees will try to manage this resource [43]. Good SSG means that employees will obtain more resources (time, knowledge, experience, etc.). After having accumulated a certain amount of resources, employees can then obtain a certain degree of autonomy and decision-making power, and superiors will then tilt work tasks to these more experienced employees, which will mean that they will not lose enthusiasm or commitment to work due to a lack of leadership [17]. On the contrary, in fact, without the constraints of superiors, subordinates will take the initiative, potentially breaking the original mode of thinking and taking proactive change behavior. In addition, high-quality SSG emphasizes the matching of values and work styles between superiors and subordinates [44], and that superiors should respect and understand the ideas of subordinates and create a humanistic and caring organizational atmosphere within the organization [17]; in this environment, employees will have more time and more energy to generate creative ideas. Therefore, we propose the following hypothesis:

**H3.** 
*Follower dependency plays a mediating role between SSG and EIB.*


### 2.4. SSG and Work Passion

Work passion is defined as the emotions and the lasting state of happiness generated by an individual based on both cognition and evaluation [45]. Individuals with high work passion tend to focus more on maintaining interpersonal relationships, and they will invest the necessary time and effort into both maintaining and improving the quality of their work relationships [46]. Indeed, previous studies have confirmed that leadership styles such as servant leadership [47] and paradoxical leadership [48] have a positive impact on work passion.

This study suggests that SSG can stimulate the work enthusiasm of employees. Firstly, trusted and cooperative SSG can provide employees with rich psychological resources, such as a sense of security and a sense of belonging, making them feel that their work is not only for the interests of the company but also for the common goals of the team and the organization as a whole, thereby enhancing both their identification with the organization and their recognition of the value of their work. Secondly, individuals with sufficient resources continuously invest in existing resources in order to obtain new resource returns [49]. Employees will constantly invest in existing resources in order to maintain or improve the quality of SSG within the current organization, and they will thereby demonstrate full enthusiasm and, concomitantly, a high level of engagement in their existing work. Finally, research has shown that leadership style [47] and high-quality LMX [50] can increase the enthusiasm of employees for work. In summary, then, we propose the following hypothesis:

**H4.** *SSG is positively associated with work passion*.

### 2.5. The Mediating Role of Work Passion

Work passion contributes to the development of EIB. People with high work enthusiasm are more inclined to explore new knowledge as well as continuously improve their cognitive level and their understanding of self-realization. When they apply their existing knowledge in order to obtain job support, this will promote their self-identity, leading to a higher willingness to innovate [51]. In addition to this, work passion can also affect employees’ emotional engagement and job satisfaction, enabling them to achieve a higher sense of self-actualization and achievement through their work. This emotion will encourage employees to demonstrate greater creativity and innovation at work, and it will, in turn, inspire them to create new values and new motivations. Finally, it has been shown that work passion can positively affect EIB [52], providing a continuous stream of positive emotional resources for employees to implement innovative activities [53]. In summary, work passion contributes to the formation of creativity.

Based on COR, this study argues that high-quality SSG can stimulate employees’ work passion and can assist them in carrying out innovative activities. On the one hand, close and good SSG can provide cognitive resources, such as psychological safety and organizational identity, to employees as a way of improving their sense of responsibility and further increasing both their recognition of the value of their work [14] and their passion for that work, which will motivate them to implement more innovative behaviors. On the other hand, according to COR, resources do not exist individually, but rather in pairs in the form of resource arrays for individuals [54]. In other words, employees who maintain a good relationship with their superiors can not only obtain relationship resources but also other resources in their favor in order to realize the gain of resources, and employees will thereby show full enthusiasm in their work, which will lead to a positive mood that will, in turn, lead to the implementation of more creative activities by employees in order to obtain more sense of achievement [53]. In summary, then, we propose the following hypothesis:

**H5.** 
*Work passion mediates the relationship between SSG and EIB.*


## 3. Methodology

### 3.1. Sample

The samples of this research come from Glad Technology Enterprises in Anhui, Zhejiang, and Jiangsu, covering various industries, including biomedical, electronic communication, and machinery manufacturing. We chose the Hi-Tech industry because studies have shown that the EIB is related to high-tech companies [7]. This research is mainly based on the online research method, and the research period was from March to May 2023. Before starting the research, the process and the purpose of the research were clarified to the human resources director of the research enterprise, and after seeking the consent of the other party, the questionnaire was distributed through wjx (a professional methodology questionnaire platform in China). In order to weaken the negative impact of common methodological bias on this study, we took several measures. First, in order to ensure that this survey is anonymous to employees, we will not disclose private personal details. Second, in the preparation of the questionnaire, the name of the variable does not appear, which ensures that the employees do not have preconceived notions, thereby ensuring, as much as possible, that they answer the question based on their true feelings. In this study, a total of 350 questionnaires were distributed and 311 questionnaires were returned, with a return rate of 88.8%. After eliminating the invalid questionnaires, such as duplicate answers and missing answers, 305 valid questionnaires were obtained, with a valid return rate of 98.07%. The percentage of men participating in the trial was 47.2%, and this was a relatively average distribution between males and females in a gender context. In terms of age, the vast majority of employees were under 45 years old, accounting for 89.2% of the workforce. Most of these people are new-generation employees, born in the 1980s and 1990s. Regarding educational background, the number of undergraduate students was the highest, with 143 students accounting for 46.9%, and there were 41 students with a master’s degree or above, which accounted for 13.5%. The proportion of undergraduate students with a bachelor’s degree or above was relatively large, which indicates that the sample generally had a higher degree level. Regarding work experience, more than 5 years accounted for 66.9%, with the vast majority of employees having worked for a longer period of time.

### 3.2. Measures

To ensure both the effectiveness and the scientific nature of the measurement tools, this study selected mature scales that have been validated in important domestic and foreign journals, and it translated some foreign language scales into Chinese according to the standard “translation back translation” method. At the same time, in order to ensure the quality of the questionnaire, we invited teachers from the English department of the school to participate in the revision of the scale. All scales used a Likert 5-point scoring system, with 1–5 indicating a range from “very agree” to “very disagree”.

SSG: This scale adopts a scale developed by Law et al. (2000) [13], with a total of 6 measurement items, such as “I often actively share or explore my ideas, problems, needs, and feelings with superiors (leaders)”. In this study, the Cronbach alpha coefficient of the scale was 0.877.

Work passion: This scale adopts the scale developed by Chen Lu et al. (2021) [16], and following the suggestion of Hair et al. (2014) [55], the items with lower factor loads were removed, ultimately retaining 8 items, such as “My work and life are very coordinated”. In this study, the Cronbach alpha coefficient of the scale was 0.930.

Follower dependency: This scale adopts the scale developed by Silke (2013) [40], and it also removes items with lower factor loads. In the end, 6 items were retained, such as “I found it difficult to carry out work normally without guidance from my direct supervisor”. In this study, the Cronbach alpha coefficient of the scale was 0.906.

EIB: This scale adopts a questionnaire developed by Zhang et al. (2016) [56], and it consists of 8 items, including “I often suggest promoting a new way of working in the company”. In this study, the Cronbach alpha coefficient of the scale was 0.946.

Control variables: In order to accurately reflect the impact of SSG on EIB, this study selected variables such as gender, age, education level, and tenure as control variables. Firstly, age and tenure have important predictive effects on EIB [57,58]. Secondly, research has confirmed that gender can also affect individuals’ evaluation of their own innovation ability [59]. Lastly, as a resource, education can be utilized by employees to implement innovative behavior [2].

## 4. Results

### 4.1. Confirmatory Factor Analysis

This study compared the four-factor model, the three-factor model, the two-factor model, and the single-factor model constructed on SSG, work passion, follower dependency, and EIB. The results are shown in Table 1. According to Table 1, the fitting degree of the four factors is better than other competitive models (χ² = 675.704; df = 344, χ²/df = 1.964, RMSER = 0.056, CFI = 0.945, TLI = 0.940, SRMR = 0.061). This indicates that the four variables used in this study have good discriminant validity.

This study further utilized the average extracted variance (AVE) and combined reliability to test the aggregated validity of the four variables. As shown in Table 2, the factor load of each measurement indicator is within a reasonable range, and the combined reliability and AVE of each variable are greater than the critical values of 0.7 and 0.5. This indicates that the four variables have good aggregated validity.

### 4.2. Common Method Bias Analysis

Firstly, the Harman single-factor test method was used to test for common method bias (CMB). Exploratory factor analysis was conducted on the variables involved in this study. The results show that four factors with eigenvalues greater than 1 were extracted and that the explanation rate of the first principal component without any rotation was 39.974%. A threshold of less than 50% indicated that no significant CMB was found. Secondly, this study also used the method of non-measurable latent factors to test for possible CMB [60]. On the basis of the four-factor model, a common factor was added to test whether the data would experience common method bias caused by measuring using the same method. After adding the common factor, the fitting results were RMSER = 0.047, CFI = 0.964, TLI = 0.957, and SRMR = 0.060. Compared with the original four-factor sub-model, the absolute values of the difference in the relevant data were 0.009, 0.019, 0.017, and 0.001. The difference between the two models was relatively small and within an acceptable range, and no significant CMB was found in the original data.

### 4.3. Descriptive Statistics and Correlation Analysis

The mean, variance, and correlation coefficients of each variable are shown in Table 3. SSG can promote EIB (r = 0.456, *p* < 0.01), is significantly negatively correlated with follower dependency (r = −0.198, *p* < 0.01), and has a significantly positive impact on work passion (r = 0.476, *p* < 0.01). Hypotheses 1, 2, and 4 were preliminarily verified.

### 4.4. Hypothesis Testing

This study used hierarchical regression for hypothesis testing, and the regression results are shown in Table 4. In Model 1, follower dependency, as the dependent variable, was measured after controlling for demographic variables, and as SSG had a significant inhibitory effect on follower dependency (b = −0.165, *p* < 0.01), Hypothesis 2 was thereby verified. In Model 2, we focused on work passion as the dependent variable, and after controlling for demographic variables, SSG could positively promote employees’ work passion (b = 0.469, *p* < 0.01), which means that Hypothesis 4 was verified. In Model 3, EIB was used as the dependent variable and, then, after adding the control variables, it was seen that the SSG could stimulate employees to implement innovative behavior (b = 0.455, *p* < 0.01), which means that Hypothesis 1 was verified, and which also indicates that SSG can stimulate EIB, thereby confirming the previous research [7]. Model 6 indicated that on the basis of Model 3, after follower dependency was added as an independent variable for regression analysis, the results showed that the positive effect of the SSG on EIB was weakened but was still significant (b = 0.41, *p* < 0.01). Model 7 indicated that on the basis of Model 3, after work passion was added as an independent variable for regression analysis, the results showed that the positive effect of SSG on EIB was weakened but was still significant (b = 0.249, *p* < 0.01), which means that subordinate dependence and work passion play a partially mediating role between SSG and EIB, and that Hypotheses 3 and 5 therefore hold.

In order to further test the mediating effect of work passion and follower dependency, this paper uses Model 4 in the PROCESS program to test the mediating effect of follower dependency and work passion on the process of SSG affecting EIB. It can be seen from Table 5 that follower dependency mediates between SSG and EIB with an effect value of 0.028 and a 95% confidence interval of [0.002, 0.067], which excludes 0, and Hypothesis 3 is thereby verified. Work passion, meanwhile, mediates between SSG and EIB with an effect value of 0.182 and a 95% confidence interval of [0.108, 0.264], which excludes 0, and Hypothesis 5 is thereby verified.

## 5. Discussion

### 5.1. Conclusions

By analyzing data from 305 full-time Chinese employees, this study draws the following conclusions. Good SSG can stimulate EIB by affecting both cognitive and affective pathways. From the cognitive perspective, good SSG can reduce follower dependency on superiors so that employees have a certain sense of autonomy and independence, which increases the possibility of employees implementing innovative forms of work behavior. From the affective perspective, good SSG can enhance the work passion of employees, and this would mean that employees have more energy and more time to engage in innovative activities.

### 5.2. Theoretical Implications

Based on COR, this study found that SSG can stimulate EIB by decreasing follower dependency and increasing the work passion of employees. This paper adds to the existing theories in various ways. First, it enriches the literature on the relationship between SSG and EIB. Compared with LMX, SSG is still in the period of theoretical development [61,62], and previous studies have found that SSG can stimulate EIB [7]. This study further supports the conclusion that SSG can stimulate EIB on the basis of previous research, which enriches the empirical research on SSG.

Second, based on COR, this study introduced follower dependency and work passion into the study of SSG, thereby enriching research on the mechanism of the role between SSG and EIB. Although previous studies have confirmed that SSG can have an impact on EIB through psychological empowerment, organizational commitment, and job satisfaction [7,12], they have neglected the role of affective factors in this type of relationship. This study confirms that the affective variable of work passion can mediate the relationship between SSG and EIB, which broadens the research perspective on the relationship between SSG and EIB.

Finally, our study enriches the research that is related to the antecedent variable of follower dependency. Previous studies have explored the effect of leadership style on follower dependency, and the empirical results have shown that a positive leadership style can increase follower dependency [17,18]. On the contrary, relative to static leadership behaviors, this study, based on COR and conducted from the Chinese context, has confirmed that the interactive behavior of good SSG can reduce follower dependency, and this answers Gu et al.’s (2016) [34] call for enhanced research on the antecedent variables of follower dependency.

### 5.3. Practical Implications

The findings of this study have important practical implications for both organizational managers and employees. On the one hand, for organizational managers, this study has confirmed that good SSG can motivate employees to implement innovative behavior, which, in turn, affects the performance of organizational innovation [26]. Therefore, managers should encourage employees to develop positive interpersonal relationships with their superiors in the daily management process rather than stopping them from doing so. For example, middle managers are encouraged to participate more or to organize group-building activities, draw closer to the relationship between themselves and their employee. In addition, managers should take the initiative to establish a benign communication channel with their subordinates while urging them to complete their work objectives, take the initiative to care for their subordinates, and provide support for subordinates and help them with their work in order to build good superior–subordinate relationships. On the other hand, for employees, good SSG means that superiors can provide more working resources so that they can always keep up the passion for working in the process of work. Employees should thus learn to maintain good interpersonal relationships with their superiors. Specifically, employees can maintain good communication with their leaders on informal occasions and can respect the power and the status of their leaders, and, at the same time, good superior–subordinate relationships should not just be a matter of asking for things, let alone relying on the superiors. In their daily work, employees should also reward their superiors for their efforts in the relationship with excellent work performance.

### 5.4. Limitations and Future Research

There are certain limitations to this study. Firstly, it only focuses on the laws at the employee level, and it does not explore the impact mechanism of SSG on EIB from multiple perspectives. Thus, the research perspective has certain limitations. Secondly, previous studies have shown that there is a significant difference between the empirical part of SSG and LMX [62], so in the future, the conceptual model in this paper can be tested with LMX, which will be more helpful to understand the difference between SSG and LMX with respect to different cultural contexts. Lastly, the present study confirms that SSG negatively affects follower dependency, whereas most of the previous studies confirm that positive leadership behaviors can increase follower dependency on superiors [17,18]. By capturing this paradox, future research can address the moderating effects of the relationship between SSG and follower dependency or a nonlinear relationship (i.e., an inverted U shape).

## Figures and Tables

**Figure 1 behavsci-13-00645-f001:**
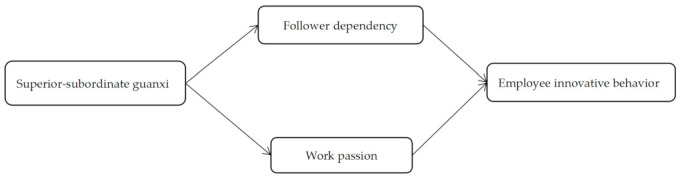
The conceptual research model.

**Table 1 behavsci-13-00645-t001:** Model fit (*n* = 305).

Model	χ²	df	χ²/df	CFI	TLI	RMSEA	SRMR
Four-factor model (SSG, FD, WP, EIB)	675.704	344	1.964	0.945	0.940	0.056	0.061
Three-factor model (SSG + FD, WP, EIB)	1483.337	347	4.275	0.812	0.796	0.104	0.152
Two-factor model (SSG + FD + WP, EIB)	2057.261	349	5.896	0.718	0.695	0.127	0.122
Single-factor model (SSG + FD + WP + EIB)	3002.148	350	8.578	0.562	0.527	0.158	0.140

Note: + = Two factors were combined; SSG = superior–subordinate guanxi; FD = follower dependency; WP = work passion; EIB = employee innovative behavior.

**Table 2 behavsci-13-00645-t002:** Results of the confirmatory factor analysis (*n* = 305).

Constructs		Loadings	CR	AVE
SSG	SSG1	0.754	0.88	0.55
	SSG2	0.751		
	SSG3	0.726		
	SSG4	0.731		
	SSG5	0.774		
	SSG6	0.691		
FD	FD1	0.703	0.91	0.62
	FD2	0.665		
	FD3	0.845		
	FD4	0.851		
	FD5	0.813		
	FD6	0.824		
WP	WP1	0.768	0.93	0.63
	WP2	0.782		
	WP3	0.848		
	WP4	0.785		
	WP5	0.831		
	WP6	0.856		
	WP7	0.748		
	WP8	0.693		
EIB	EIB1	0.852	0.95	0.69
	EIB2	0.89		
	EIB3	0.836		
	EIB4	0.85		
	EIB5	0.806		
	EIB6	0.712		
	EIB7	0.839		
	EIB8	0.843		

Note: SSG = superior–subordinate guanxi; FD = follower dependency; WP = work passion; EIB = employee innovative behavior.

**Table 3 behavsci-13-00645-t003:** Mean, standard deviation, and correlation coefficient of variables (*n* = 305).

Variables	M	SD	1	2	3	4	5	6	7
1. gender	1.53	0.50							
2. age	2.96	1.26	0.088						
3. edu	2.55	0.97	−0.123 *	−0.282 **					
4. tenure	3.27	1.41	0.02	0.824 **	−0.196 **				
5. SSG	3.08	0.86	−0.012	0.099	−0.032	0.134 *			
6. FD	2.60	0.92	−0.121 *	−0.265 **	0.225 **	−0.272 **	−0.198 **		
7. WP	3.55	0.86	0.068	0.154 **	−0.162 **	0.123 *	0.476 **	−0.385 **	
8. EIB	3.50	0.87	−0.019	0.081	−0.146 *	0.061	0.456 **	−0.349 **	0.563 **

Note: * *p* < 0.05, ** *p* < 0.01; SSG = superior–subordinate guanxi; FD = follower dependency; WP = work passion; EIB = employee innovative behavior.

**Table 4 behavsci-13-00645-t004:** Summary of the hierarchical regression results (*n* = 305).

Variables	FD	WP	EIB				
	Model 1	Model 2	Model 3	Model 4	Model 5	Model 6	Model 7
Gender	−0.096	0.049	−0.035	−0.074	−0.065	−0.061	−0.056
Age	−0.05	0.123	0.07	0.038	−0.006	0.056	0.016
Edu	0.16	−0.119	−0.132	−0.081	−0.068	−0.088	−0.08
Tenure	−0.175	−0.066	−0.083	−0.081	−0.015	−0.131	−0.051
SSG	−0.165 **	0.469 **	0.455 **			0.41 **	0.249 **
FD				−0.352 **		−0.277 **	
WP					0.56 **		0.44 **
R^2^	0.14	0.26	0.23	0.13	0.33	0.29	0.37
F	9.83 **	20.67 **	17.68 **	9.27 **	28.8 **	20.67 **	29.44 **

Note: ** *p* < 0.01; SSG = superior–subordinate guanxi, FD = follower dependency, WP = work passion, EIB = employee innovative behavior.

**Table 5 behavsci-13-00645-t005:** Summary of the mediation analysis results (*n* = 305).

Relationship	Effect	SE	BootLLCI	BootULCI
Total effect	0.459	0.052	0.352	0.527
Direct effect	0.249	0.057	0.139	0.359
Total indirect effect	0.210	0.041	0.131	0.292
FD	0.028	0.017	0.002	0.067
WP	0.182	0.040	0.108	0.264

Note: FD = follower dependency, WP = work passion.

## Data Availability

Data collected and analyzed during the study are available upon reasonable request.

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
