# Peer review of "Effects of Superior–Subordinate Guanxi on Employee Innovative Behavior: The Role of Follower Dependency and Work Passion"

_behavsci, 2023, doi:10.3390/bs13080645_

Round 1
Reviewer 1 Report
1. What is the main question addressed by the research?
This study investigates the mechanism of the role of superior-subordinate relationships on employees' innovative behaviour from the emotional and cognitive perspectives
2. Do you consider the topic original or relevant in the field? Does it address a specific gap in the field?
In the Asian context, the phenomenon provides knowledge, in addition to approaching it from the emotional and cognitive perspectives.
3. What does it add to the subject area compared with other published material?
Precisely the methodological approach from the emotional and cognitive perspectives, generates a more comprehensive picture of the behavior of employees.
4. What specific improvements should the authors consider regarding the methodology? What further controls should be considered?
Methodologically, perhaps it is relevant to explain in detail how many, which and what type of companies the employees who responded work for. "The questionnaire link was distributed to surrounding full-time employees through chat software such as WeChat and DingTalk" When?
5. Are the conclusions consistent with the evidence and arguments presented and do they address the main question posed?
YES
6. Are the references appropriate?
Yes
7. The tables and figures are presented according to the standards
Specify in the discussion part with which studies you agree and differ, in relation to the results. Replace the mentions of previous studies, other studies, by the above.
Reviewer 2 Report
The manuscript focuses on one of the contemporary issues, which I think is very interesting and a complex issue too. The authors have managed to articulate the issue reasonably well in the introduction.
Introduction part can have a subsection about the research problem and can also show the research questions clearly. More clarity on the term “Guanxi” will be needed in the introduction section, as readers who are not culturally aware of the term would find difficulty in relating the concept.
The study needs to written gender neutrally Eg. See “he’ in Line 52.
The conceptual model can be aligned with the Literature review and the proposed hypotheses. Conceptual framework has “Superior-subordinate guanxi” and “Employee innovative behaviour”, however only “follower dependency” and “work passion” reviews are provided. Please provide literature reviews for “Superior-subordinate guanxi” and “Employee innovative behaviour”. Also, hypotheses for “Superior-subordinate guanxi” and “Employee innovative behaviour” can be proposed and tested. Also, ensure that latest literature is provided.
Methodologically, the paper does not lack rigour, but it needs more clarity on the sample selected, demographics and also potential limitations due to the methodology. It does not mention about the “representativeness of the sample”. Methodology section need to also include details about the research design, research philosophy, research approach, etc. Provide a clear and detailed explanation of the research design and methodology employed in the study. Clearly state the research approach and justify its suitability for addressing the research questions. Also, it is not clearly mentioned whether the research instrument has been validated Eg. Content Validity, Construct Validity, etc.
Results section should go beyond a mere summary and interpretation of the findings. Provide an in-depth analysis of the data, relating it back to the research questions and objectives. Clearly explain the implications of the findings and their significance in the context of prevailing literature available. Additionally, discuss any unexpected or contradictory findings and offer potential explanations or avenues for further research.
The discussion section should include a comprehensive review and synthesis of relevant literature. Compare the findings of the study with existing research in the research area. Identify areas of agreement or divergence and discuss the potential reasons behind these differences. This will strengthen the validity and generalizability of the study's findings. The manuscript’s “5.2. Theoretical implications” has only made some attempt to relate with prevailing literature. Rest of the sections in “Section 5 Discussion” does not have any attempted to relate with the literature.
Review the manuscript for overall coherence and flow, ensuring that the content is well-organized and logically structured.
Based on the review, I recommend the manuscript to be reconsidered after the revisions suggested.
The study needs to written gender neutrally Eg. See “he’ in Line 52.
Reviewer 3 Report
This paper seems to be an interesting research topic. The purpose of the study was clear, so it was readable.
However, the logic of the hypotheses needs to be clearer. For example, unlike previous studies, you assumed that the relationship between SSG and follower dependence would be negative since a good SSG can help employees especially the new generation of employees no longer adheres to the thoughts and opinions of leaders. However, in the actual analysis, age was used only as a control variable.
Furthermore, since the mediating effect provides important implications for this paper, the logic of the mediating effect should be revealed in more detail. In the current paper, the mechanism is too simple.
Finally, it is recommended that future studies address the moderating effects of the relationship between SSG and follower dependence or a non-linear relationship(i.e. inverted U shape).
* It's trivial, but the reference number is wrong, so please correct it (p.10 #5~#1, p.11 #19~#2)
Thank you for your hard work in carrying out the research.
Round 2
Reviewer 2 Report
As the comments have been addressed, the manuscript may be published. Formatting and minor edits for language need to be done.
Minor edits for language need to be made.